# COVID-19 Detection Empowered with Machine Learning and Deep Learning Techniques: A Systematic Review

**Amir Rehman, Muhammad Azhar Iqbal, Huanlai Xing *** and **Irfan Ahmed**

School of Computing and Artificial Intelligence, Southwest Jiaotong University, Chengdu 611756, China; amir@my.swjtu.edu.cn (A.R.); m.a.iqbal@swjtu.edu.cn (M.A.I.); irfanabbasi@my.swjtu.edu.cn (I.A.)
* Correspondence: hxx@home.swjtu.edu.cn; Tel.: +86-187-8027-2961

**Abstract:** COVID-19 has infected 223 countries and caused 2.8 million deaths worldwide (at the time of writing this article), and the death rate is increasing continuously. Early diagnosis of COVID patients is a critical challenge for medical practitioners, governments, organizations, and countries to overcome the rapid spread of the deadly virus in any geographical area. In this situation, the previous epidemic evidence on Machine Learning (ML) and Deep Learning (DL) techniques encouraged the researchers to play a significant role in detecting COVID-19. Similarly, the rising scope of ML/DL methodologies in the medical domain also advocates its significant role in COVID-19 detection. This systematic review presents ML and DL techniques practiced in this era to predict, diagnose, classify, and detect the coronavirus. In this study, the data was retrieved from three prevalent full-text archives, i.e., Science Direct, Web of Science, and PubMed, using the search code strategy on 16 March 2021. Using professional assessment, among 961 articles retrieved by an initial query, only 40 articles focusing on ML/DL-based COVID-19 detection schemes were selected. Findings have been presented as a country-wise distribution of publications, article frequency, various data collection, analyzed datasets, sample sizes, and applied ML/DL techniques. Precisely, this study reveals that ML/DL technique accuracy lay between 80% to 100% when detecting COVID-19. The RT-PCR-based model with Support Vector Machine (SVM) exhibited the lowest accuracy (80%), whereas the X-ray-based model achieved the highest accuracy (99.7%) using a deep convolutional neural network. However, current studies have shown that an anal swab test is super accurate to detect the virus. Moreover, this review addresses the limitations of COVID-19 detection along with the detailed discussion of the prevailing challenges and future research directions, which eventually highlight outstanding issues.

**Keywords:** COVID-19; detection; machine learning; deep learning; diagnosis; anal swab; systematic review; computer-aided diagnosis

## 1. Introduction

COVID-19 is a contagious disease reverted by the SARS virus and announced as a global pandemic by the World Health Organization (WHO) in March 2020. The COVID-19 massive outbreak has infected 223 countries, with more than 129 million infected patients and 2.8 million deaths globally [1,2]. The infected cases and death rate is increasing rapidly, as shown in Figure 1. The vital factor is to diagnose the infected cases at an early stage to control this natural pandemic. However, advanced intelligent prediction systems and accurate modeling of techniques have made a precious contribution in managing and planning health resources against the virus. The initial stage diagnosis of the virus is also useful for appropriate patient isolation, fast canulization of chronic patients in specific hospitals, and observing virus spread. However, the diagnosis of COVID-19 is quite challenging given the economic issues raised among developed and under-developed countries due to the high cost of diagnostic tests [3].

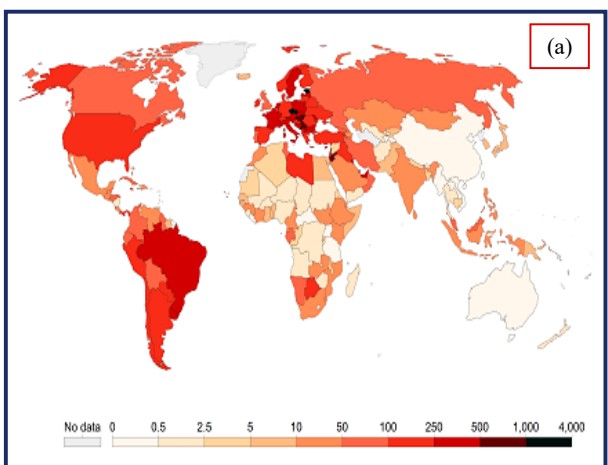 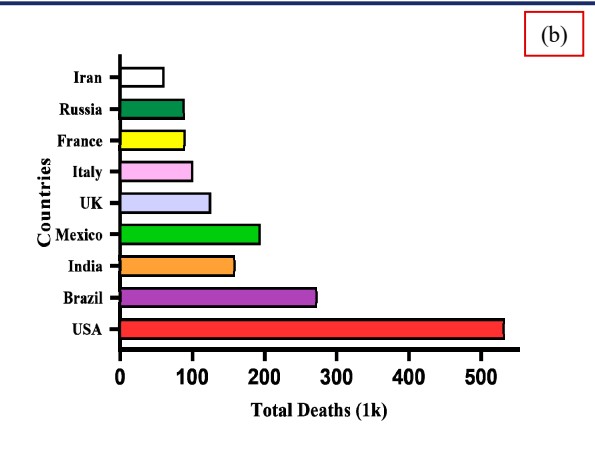

**Figure 1.** (**a**) Daily new confirmed COVID-19 cases Worldwide (**b**). COVID-19 death rate statistics among top ten infected countries last updated 17 March 2021 [2]. (Sources: Center for Systems Science and Engineering (C.S.S.E.) at Johns Hopkins University, Baltimore, MD, USA).

One of the most critical reasons behind the rapid spread of COVID-19 is the shortage of clinical diagnosis methods [4], such as reverse transcription-polymerase chain reaction (RT-PCR) [5], serologic tests, and viral throat swab testing (oropharyngeal) [6], etc. Currently, China has started anal swab testing to enhance the detection rate and overcome the transmission of the deadly virus. Researchers have found that anal swab testing accuracy is higher than other diagnostic swab methods [7], but this method is still not convenient for massive sampling. These methods are quite expensive and not readily available in many countries and have been used only to detect the existence of the virus [8]. Therefore, researchers proposed some other techniques such as Computed Tomography (C.T.) scans and X-ray to assess the screening and intensity of COVID-19 infection (in hospital admitted cases). These tests are more suitable for providing fast and accurate results [9]. It is still challenging to use chest C.T. scans and chest radiographs (X-rays) because they may not correctly distinguish between COVID-19 infection and other lung diseases [10,11].

However, the above-mentioned conventional methods require medical experts for the judgment of diagnostic tests. To automate this type of diagnosis, a computer-aided diagnosis (CAD) system provides vital solutions to researchers detecting coronavirus by using classification approaches [12]. CAD has now become the center of research. In the case of COVID-19, CAD employs ML/DL techniques to evaluate all kinds of patient data, either images or clinical data, and predict the patient's condition. This evaluation improves diagnosis and support for medical experts in decision making and improves diagnosis [13]. Henceforward, the CAD system is considered a robust tool for radiologists to improve diagnostic methods such as chest radiographs, C.T. Images, etc. Thus, ML and DL techniques have been introduced to develop expert systems due to their significant role in healthcare to deploy clinical decision support systems [14]. ML and DL are not only supportive for the classification of medical diseases such as breast cancer [15], tuberculosis [16], pneumonia [17], and muscle diseases [18] but also useful for detecting, diagnosing, and classifying COVID-19 [19,20]. ML and DL are emerging technologies in the field of medicine, and due to their capability to generate highly accurate predictive results, these methods can play an essential role in renovating the healthcare system [21]. However, due to the enormous variety of these techniques, it is difficult to prefer an adequate ML/DL strategy that provides effective results [22,23] to diagnose COVID-19.

Moreover, most of these techniques are characterized by less accuracy and computational efficiency [24]. Due to these issues, it is a big challenge to evaluate the optimal technique having the most accurate results [25]. In other words, it is very crucial to present a prediction system that can precisely detect and diagnose the virus. It will also be a challenge for ML techniques to detect public health risks in advance for improving the

prediction, prevention, and detection of future epidemiologic risks. In literature, several ML/DL techniques have been proposed as intelligent systems to provide highly accurate results in a short time for the prediction of COVID-19 [26–28]. In this review, our primary focus is to accomplish the following two tasks: (i) the analysis of different ML/DL techniques that have been recently published related to the detection of coronavirus and (ii) consideration of upcoming research challenges. Furthermore, novel efforts have been made to achieve the following objectives:

➢   Determining the significance of available datasets from literature used for the prediction of COVID-19.
➢   To analyze the ML and DL techniques that were applied to detect the COVID-19.
➢   Identification of challenges and future research directions related to the implications of ML/DL techniques for COVID detection.

The study can be categorized as follows. Section 2 describes the procedures used to evaluate the systematic literature on the topic. Results and relevant discussion are explained in Section 3. Then introduce the anal swab testing in Section 4. Section 5 discusses the upcoming challenges and research directions of ML/DL related to COVID-19. In the end, the conclusions and future directions of the study are presented.

## 2. Review Methodology

The methodology for the current study was adopted from the Preferred Reporting Items for Systematic Reviews and Meta-Analyses (P.R.I.S.M.A.) statement [29].

### 2.1. Selection of Digital Archives

For this systematic review, the data was collected from three full-text digital archives, including Science Direct (S.D.), Web of Science (WoS), and PubMed. Google Scholar and medRxiv were employed for preprint studies. S.D. provides access to exceptionally reliable journals in the field of engineering, medical, and computer technology. WoS is a comprehensive source of social science, liberal arts, and multi-disciplinary applications. PubMed accesses MEDLINE, which is the most suitable database for medical and bio-engineering studies [30]. Google scholar is a reliable source in every research field, such as biomedical engineering, computer science, and health technology. These databases were selected due to their adequate scope and originality of the studies. All hypothetical features of coronavirus are included in these databases as a diagnosis of the virus needs more attention than other infections.

### 2.2. Search Code Strategy

The original research articles available from the beginning of time were collected from the databases (S.D., WoS, PubMed) on 16 March 2021. Many queries were used to enhance the search-related quality to various ML and DL techniques for detecting COVID-19. Important searching keywords used for the data retrieval were as follows: (a) SD database search code is TOPIC: (("Coronavirus" OR "COVID-19") AND ("Detection" OR "diagnosis" OR "classification") AND ("Machine learning" OR "Deep learning")); (b) PubMed database search code is TOPIC: (("Novel Coronavirus" OR "COVID-19" OR "coronavirus" OR "2019-nCoV" OR "new coronavirus" OR "SARS-CoV-2") AND ("Detection" OR "diagnosis" OR "prediction" OR "prognosis") AND ("Machine learning" OR "Deep learning"));(c) WoS database search code is TOPIC:("Novel Coronavirus" OR "COVID-19" OR "coronavirus" OR "2019-nCoV" OR "new coronavirus" OR "SARS-CoV-2") AND ("Detection" OR "diagnosis" OR "prediction" OR "prognosis" OR "Analysis" OR "Classification") AND ("Machine learning" OR "Deep learning"). The published papers with these terms found in titles, abstracts, and keywords were retrieved. Moreover, only original research papers related to the detection of COVID-19 through ML/DL techniques were extracted, and the review articles and books were eliminated from the collection.

Machine and Deep Learning Techniques

Remarkable progress has been made in using different ML algorithms applied to medical datasets for detecting different diseases. ML is a mathematical and statistical technique that gives computers the ability to learn from data and elaborate a convoluted framework. DL consists of a group of algorithms applied to develop an expert system that can identify problems and yield predictions. These ML and DL techniques produce intelligence into a computer that can elicit the patterns relative to specific data and then proceeds for automatic reasoning [31]. Many ML/DL algorithms have been applied to predict, detect, and diagnose COVID-19 [32].

In this article, we have presented an analysis of these ML/DL techniques used to detect COVID-19. Below is a brief description of these classifiers.

SVM is a supervised ML classifier, a specified set of training examples given to SVM that learn the hyperplane, isolate the instances from each class, and magnify the edges among data instances and hyperplane. K-NN is also a supervised algorithm. It classifies the unknown sample by measuring its distance from other training samples and calculates K-smallest distances. The unknown sample output class label is given by most represented in these K classes [33].

A Decision Tree classifier consists of one root node, numerous internal nodes, and several terminal nodes. D.T. represents a tree structure in which every leaf node is related to a group of features, and a branch displays a value. The primary purpose of the D.T. algorithm is to build a tree for the whole data and provide refine results on each leaf. A Random Forest classifier ensembles various decision trees for prediction, which are developed by grasping numerous classification trees simultaneously R.F. performance is more superior to a single tree [34]. This classifier can be employed efficiently, and it accomplishes predictions for many higher dimensional datasets with greater accuracy.

CNN and D.N.N. is a deep learning technique consisting of several layers such as input, output, and hidden layers. These layers modify the data with convolution filters. Currently, researchers are employing these deep learning techniques to detect COVID-19 [31].

It is investigated in the literature that these models achieved greater accuracy in detecting viral pneumonia, bacterial pneumonia, and COVID-19 [26], respectively. ML/DL represents a tremendous innovation in automatic diagnostic classification systems; therefore, these classifiers present a suitable choice to be employed in diagnosing COVID-19.

### 2.3. Eligibility Criteria and Article Screening

The literature selection method was started by applying search code queries on selected three digital databases, as shown in Figure 2. In the initial query, 961 articles were found; however, by removing duplicate articles from these, 886 articles have remained. Screening has been conducted based on title, abstract, and keywords, then 795 articles were excluded. We had sorted out 91 articles for complete text studies, and after studying each article, only 40 high-quality articles were selected for this systematic review.

### 2.4. Data Segregation and Categorization

The data were extracted and categorized to investigate the adaptability of the COVID in terms of exposure, diagnosis, and classification through ML/DL techniques. The distribution of the retrieved data with the countries involved in the study of COVID-19 detection through ML/DL techniques is presented in Figure 3.

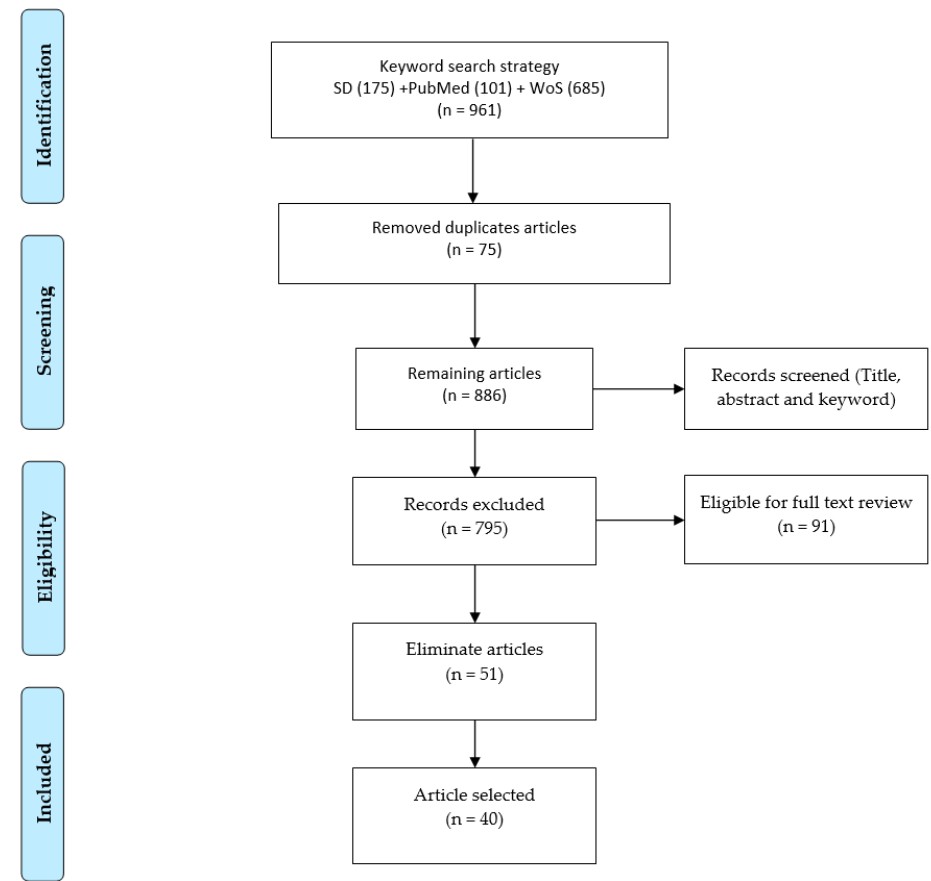

**Figure 2.** Search and selection strategy for systematic review.

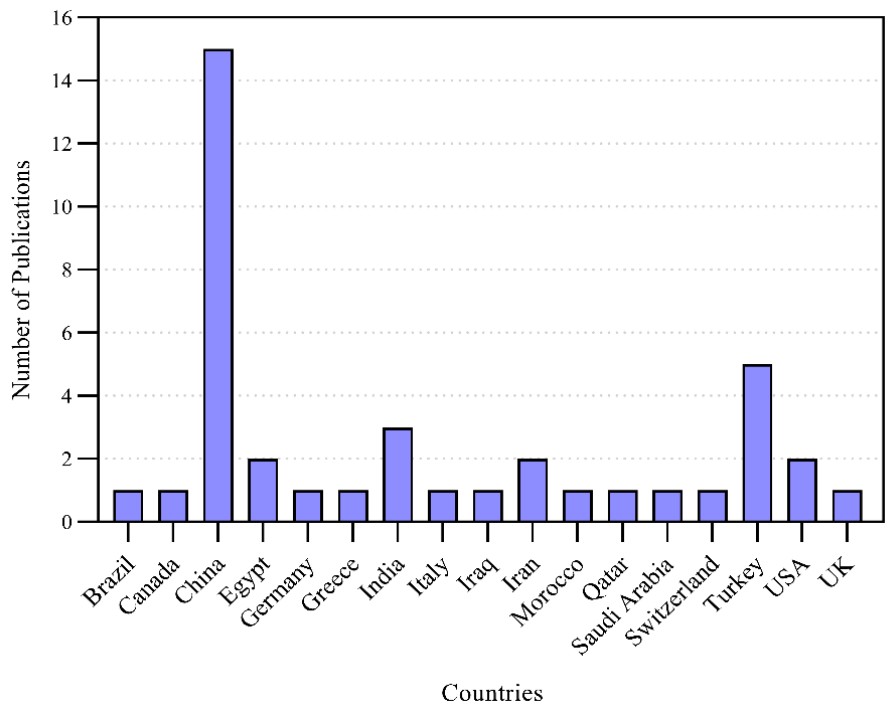

**Figure 3.** Distribution of published articles (ML/DL techniques for COVID-19 prediction).

It can be noticed that China was the most prolific country with 15 publications related to the MLT-based detection of COVID-19. Some other influential contributing countries (and their publications) were Turkey (5), India (3), USA (2), Iran (2), and Egypt (2). The rising scope of the ML/DL techniques can play a significant role in controlling the spread of the novel coronavirus globally. Furthermore, different ML and DL algorithms such as classification, regression, and prognosis were presented to analyze these technique's efficiency. Figure 4. highlights the number of articles related to these algorithms utilized for the detection of COVID-19. It can be observed that most of the studies employed Convolutional Neural Network (CNN) algorithm and Deep Neural Network (D.N.N.) algorithm. The utilization frequencies of these techniques were 18 and 13, respectively. Only five investigations used Support Vector Machine (SVM) and Random Forest (R.F.) algorithms to detect the COVID, whereas the number of investigations related to Long Short-Term Memory (L.S.T.M.), Decision Tree (D.T.), and K-Nearest Neighbors (K-NN) algorithms was 4, 2, and 1, accordingly.

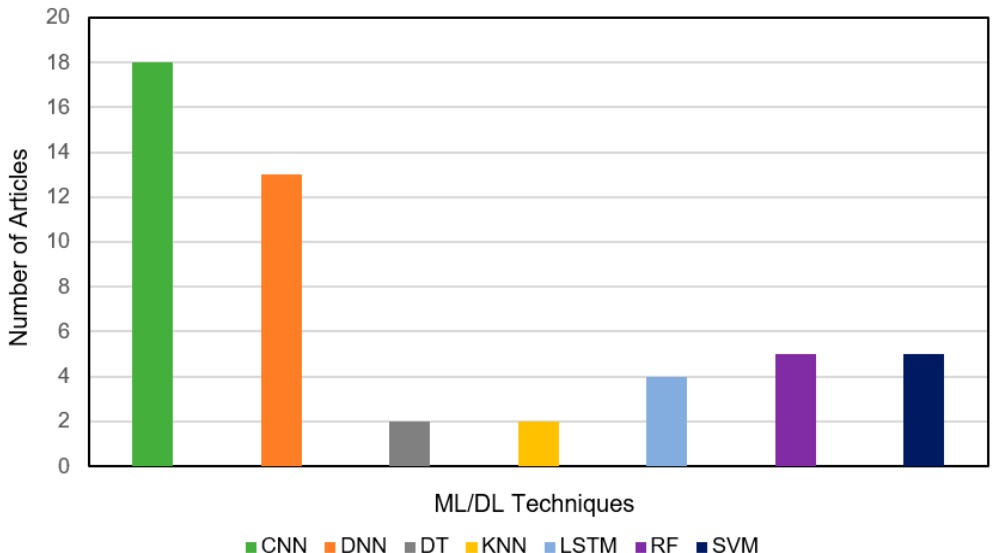

**Figure 4.** ML/DL techniques used in this review articles.

The research articles retrieved for the current systematic review and the available datasets that used ML/DL techniques (such as SVM, K-NN, R.F., CNN, and D.N.N.) for the diagnosis of COVID-19 elaborate in Section 3.

## 3. Results and Discussion

### 3.1. Analysis of Extracted Data

Table 1 summarizes the datasets used in the previous studies presented along with test data (i.e., Chest X-rays and C.T. Images, RT-PCR, and Clinical data), types of infected patients, and references. In some articles, public databases such as Kaggle and GitHub repository were utilized, whereas, in other studies, the private data collected from hospitals and universities were investigated. The diagnosis of COVID-19 is crucial as careful investigations and data processing are required using specific medical expertise. It was revealed that in most of the cases, the data from C.T. and X-ray images were studied, whereas a few investigations were conducted using RT-PCR and clinical blood test data to diagnose COVID-19 by using ML/DL techniques.

**Table 1.** Summary of the datasets used in the selected papers.

| Reference | Test Data | COVID-19 Positive | Pneumonia/Other Infection | Healthy | Total | Sources/Links |
|---|---|---|---|---|---|---|
| [35] | Chest X-ray CT Images | 50 | * NA | 50 | 100 | https://github.com/ieee8023/covid-chestxray-dataset https://www.kaggle.com/paultimothymooney/chest-xray |
| [36] | Chest X-ray | 284 | 657 | 310 | 1251 | https://github.com/ieee8023/covid-chestxray-dataset https://www.kaggle.com/paultimothymooney/chest-xray |
| [37] | Chest X-ray | 250 | 2753 | 3520 | 6523 | https://github.com/ieee8023/covid-chestxray-dataset https://github.com/muhammedtalo/COVID-19 https://www.kaggle.com/nih-chest-xrays/sample |
| [38] | Chest X-ray | 274 | 2051 | 1341 | 3666 | https://www.kaggle.com/tawsifurrahman/covid19-radiography-database https://github.com/agchung/Figure1-COVID-chestxray-dataset https://github.com/ieee8023/covid-chestxray-dataset |
| [26] | CT Images | 1296 | 1735 | 1325 | 4356 | https://github.com/bkong999/COVNet |
| [39] | CT Images | 510 | 510 | * NA | 1020 | Alexion, Toshiba Medical System, Japan |
| [11] | Chest X-ray | 250 | 500 | 1000 | 1750 | https://github.com/ieee8023/COVID-chestxray-dataset |
| [40] | CT Images | 219 | 224 | 175 | 618 | Hospital of Zhejiang University Hospital of Wenzhou Hospital of Wenling |
| [41] | CT Images | 313 | * NA | 229 | 542 | Union Hospital Tongji Medical College, Huazhong University of Science and Technology, China. |
| [42] | CT Images | 325 | 740 | * NA | 1065 | Xi'an Jiaotong University First Affiliated Hospital Nanchang University First Hospital Xi'an No.8 Hospital of Xi'an Medical College, China |
| [43] | CT Images | 777 | ** NS | 708 | 1485 | The Third Affiliated Hospital and Sun Yat-Sen Memorial Hospital, Sun Yat-sen University Guangzhou Renmin Hospital of Wuhan University, China |

**Table 1.** *Cont.*

| Reference | Test Data | COVID-19 Positive | Pneumonia/Other Infection | Healthy | Total | Sources/Links |
|---|---|---|---|---|---|---|
| [44] | CT Images | 306 | * NA | 306 | 612 | University of Medical Science (I.U.M.S.), Iran |
| [3] | Chest X-ray | 224 | 700 | 504 | 1428 | https://github.com/ieee8023/covid-chestxray-dataset https://www.kaggle.com/andrewmvd/convid19-X-rays |
| [45] | Chest X-ray | 53 | 5526 | 8066 | 13,645 | https://github.com/agchung/Figure1-COVID-chestxray-dataset https://github.com/agchung/Actualmed-COVID-chestxray-dataset https://www.kaggle.com/tawsifurrahman/covid19-radiography-database https://www.kaggle.com/c/rsna-pneumonia-detection-challenge/data |
| [46] | Chest X-ray | 127 | 127 | 127 | 381 | https://openi.nlm.nih.gov/ www.kaggle.com www.github.com |
| [47] | Chest X-ray | 25 | * NA | 25 | 50 | https://github.com/ieee8023/covid-chestxray-dataset https://www.pyimagesearch.com/category/medical/ |
| [48] | Chest X-ray | 239 + 2265 (BIMCV) | 4273 + 951 | 1583 | 9311 | Japanese Radiological Scientific Technology (J.R.S.T.) Shenzhen Dataset, Montgomery Dataset University of Montreal, Valencian Region Medical Image Bank (B.I.M.C.V.) https://github.com/ieee8023/covid-chestxray-dataset |
| [49] | CT Images | 56 | 52 | 49 | 157 | http://www.chainz.cn/Hospital in Wenzhou, ChinaEI-Camino Hospital, USA |
| [50] | CT Images (abdominal) | 53 | ** NS | * NA | 150 | https://www.sirm.org/en/ |
| [51] | Chest X-ray | 68 | 2786 | 1583+ 1504 | 5941 | https://github.com/ieee8023/covid-chestxray-dataset |

**Table 1.** *Cont.*

| Reference | Test Data | COVID-19 Positive | Pneumonia/Other Infection | Healthy | Total | Sources/Links |
|-----------|-----------|-------------------|---------------------------|---------|-------|---------------|
| [52] | CT Images | 133 | ** N.S. | * NA | 199 | Wuhan Pulmonary Hospital, china |
| [53] | Chest X-ray andCT Images | 2228 | 3308 | 2381 | 7917 | https://github.com/ChenWWWeixiang/diagnosis_covid19 https://tianchi.aliyun.com/competition/entrance/231601/information Wuhan Union Hospital and Jianghan Mobile cabin Hospital, China |
| [54] | CT Images | 877 | * NA | 541 | 1418 | Beijing Tsinghua Changgung Hospital, China, Wuhan No.7 Hospital, Wuhan Leishenshan Hospital, Zhongnan Hospital of Wuhan University, Wuhan, Tianyou Hospital Affiliated to Wuhan University of Science and Technology, Wuhan, China |
| [55] | Chest X-ray | 423 | 1485 | 1579 | 3487 | https://www.kaggle.com/tawsifurrahman/covid19-radiography-database https://www.sirm.org/category/senza-categoria/covid-19/ https://github.com/ieee8023/covid-chestxray-dataset https://www.kaggle.com/paultimothymooney/chest-xray-pneumonia |
| [56] | Chest X-ray and CT Images | 288 | ** NS | 238 | 526 | https://www.bsti.org.uk/training-and-education/covid-19-bsti-imaging-database/ https://radiopaedia.org/articles/normal-chest-imaging-examples?lang=gb https://www.kaggle.com/paultimothymooney/chest-xray-pneumonia/metadata |
| [57] | Chest X-ray | 100 | ** NS | 1431 | 1531 | https://github.com/ieee8023/covid-chestxray-dataset |
| [58] | RT-PCR | 102 | ** NS | * NA | 235 | Hospital Israelita Albert Einstein, Brazil |
| [59] | RT-PCR | ** NS | ** NS | * NA | 53 | Wenzhou Central Hospital and Cangnan People's Hospital Wenzhou, China |

**Table 1.** *Cont.*

| Reference | Test Data | COVID-19 Positive | Pneumonia/Other Infection | Healthy | Total | Sources/Links |
|---|---|---|---|---|---|---|
| [60] | Chest CT Images | 924 | 4448 | * NA | 5372 | Renmin Hospital of Wuhan University, Henan Provincial People's hospital, First Hospital of China Medical University First Affiliated Hospital of Anhui Medical University Beijing Youan Hospital of Capital Medical University, Huangshi Central Hospital, |
| [61] | Chest CT Images | 349 | * NA | 463 | 812 | https://www.sirm.org/ https://radiopaedia.org/ https://www.kaggle.com/ tawsifurrahman/covid19-radiography-database/ https://coronacases.org/ https://www.eurorad.org/ |
| [62] | Chest X-ray CT Images | ** NS | 4273 | 1583 | 5856 | [63] |
| [64] | Chest X-ray | 45 | 931 | 660 | 1636 | https://github.com/ lindawangg/COVID-Net https://www.kaggle.com/ c/rsna-pneumonia-detection-challenge/data |
| [65] | Chest X-ray | 105 | * NA | 80 | 185 | https://github.com/ieee8023/covid-chestxray-dataset |
| [66] | Chest X-ray | 680 | 1845 + 3457 | 9977 | 15,959 | https://www.kaggle.com/ c/rsna-pneumonia-detection-challenge https://www.kaggle.com/ tawsifurrahman/covid19-radiography-database https://github.com/ieee8023/covid-chestxray-dataset https://github.com/ muhammedtalo/COVID-19 |
| [67] | Chest X-ray | 534 | 1157 | 1310 | 3001 | https://github.com/ieee8023/covid-chestxray-dataset https://www.kaggle.com/ paultimothymooney/chest-Xray |
| [68] | Chest X-ray | 210 | 350 | 350 | 910 | https://www.kaggle.com/ tawsifurrahman/covid19-radiography-database |
| [69] | Chest X-ray | 696 | 696 | 696 | 2088 | https://github.com/ieee8023/covid-chestxray-dataset |
| [70] | Clinical Blood Test | Suspected Covid-19 105 | 148 | * NA | 253 | Gansu Provincial Hospital, Lanzhou Pulmonary Hospital, The First Hospital of Lanzhou University, The First People's Hospital of Lanzhou City, Lanzhou University Second Hospital, China |

**Table 1.** *Cont.*

| Reference | Test Data | COVID-19 Positive | Pneumonia/Other Infection | Healthy | Total | Sources/Links |
|---|---|---|---|---|---|---|
| [71] | Clinical Blood Test | 160 | 5333 | * NA | 5493 | University Medical Centre Ljubljana (U.M.C.L.), Slovenia |
| [72] | Clinical Blood Test | 82 | * NA | 32 | 114 | Taizhou Hospital Zhejiang, China |

* Not Applied (N.A.) represents those kinds of images that were not utilized; ** Not Specified (N.S.) shows the images that were utilized but did not explain the number clearly.In Table 1, some significant available data sources were also presented and utilized by other researchers to explore more valuable solutions to deal with this pandemic. The datasets using X-ray images were higher in number than those with C.T. images. The percentage contribution of different diagnostic methods used for COVID-19 detection is presented in Figure 5. It can be noticed that X-ray image-based detection was the most utilized diagnostic method, contributing 47% of the total. C.T. image-based models were the second-largest diagnostic methods accounting for 30%, whereas the approaches with the least utilization for diagnosis of the virus were X-ray and C.T. image-based combined models (10%), clinical blood test (8%), and RT-PCR data-based models (5%).

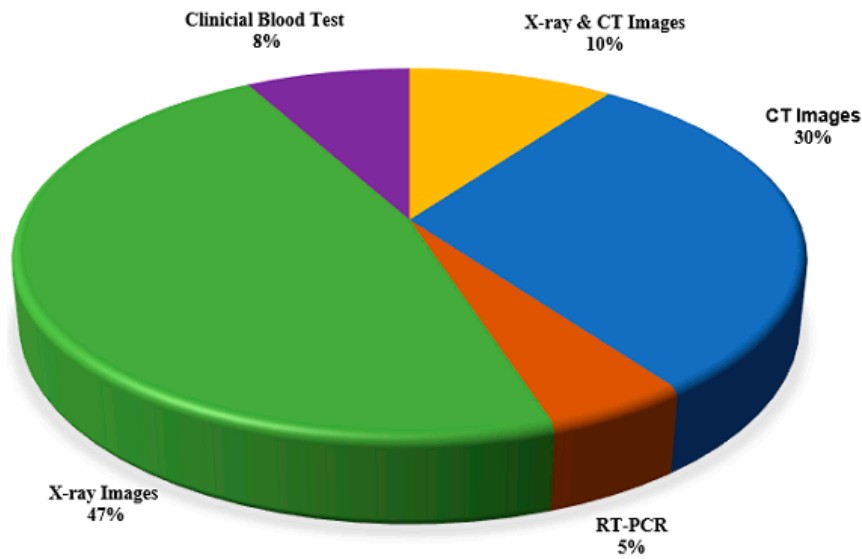

**Figure 5.** Percentage utilization of various diagnostic techniques for the detection of COVID-19.

Generally, there are two methods of prediction of COVID-19 images, binary classification and multiclass classification. Binary classification is used to detect COVID-19 positive and negative cases. However, this classification method is inaccurate due to the misclassification of COVID-19 images with other types of lung diseases (viral pneumonia, bacterial pneumonia). To solve this issue, researchers have differentiated the images of COVID-19 viral pneumonia, bacterial pneumonia, fungal pneumonia, SARS, MERS, influenza, tuberculosis, and images of healthy people by classifying them using the method of multiclass classification. The accuracy of multiple classifiers is better than binary classifiers in detecting COVID-19 images. The total number of C.T. scans and X-ray images used to categorize the COVID Positive, Normal, Pneumonia, and other Lung patients were reported in the literature and are mentioned in Table 1.

### 3.2. Investigation on Classification Performance

Table 2 summarizes the state-of-art COVID-19 prediction algorithms to test the data (Radiological (chest X-rays and C.T. images), RT-PCR, and Clinical data) along with the highest prediction results of the selected previous studies. Only the best-obtained results of different ML/DL techniques on C.T., X-ray images, RT-PCR, and clinical blood test data were mentioned.

**Table 2.** Summary of state-of-art COVID-19 prediction algorithms.

| Reference | Test Type | ML/DL Techniques | Prediction Results | Country | Cited by No of Papers |
|---|---|---|---|---|---|
| [26] | CT Images | CNN, COVNet | AUC 0.96 | China | 553 |
| [39] | CT Images | CNNs, ResNet-101 & Xception | AUC of 0.99 Sensitivity 98.02% Specificity 99.51% | Iran | 120 |
| [41] | CT Images | 3-D DNN, DeCoVNet | Accuracy 0.90 | China | 205 |
| [42] | C.T. Images | Inception Transfer learning model establish the algorithm | Accuracy of 89.5% with Specificity of 0.88 and Sensitivity of 0.87 | China | 376 |
| [43] | C.T. Images | D.N.N., DRE-Net | A.U.C. of 0.99 Sensitivity of 0.93 | China | 198 |
| [49] | CT Images | 2D and 3D deep learning (Resnet-50-2D) and AI Models | AUC of 0.99 Sensitive 92.2% Specificity 92.2% | China | 306 |
| [50] | C.T. Images | Classification Stage 1 SVM, Stage 2 GLCM, GLSZ MDWT | Accuracy of 99.68% | Turkey | 103 |
| [52] | C.T. Images | Multilayer perceptron and LSTM | AUC of 0.954 | China | 48 |
| [54] | CT Images | Combined model 3D UNet++ and RestNet-50 | AUC of 0.991 Sensitivity of 0.974 and specificity of 0.922 | China | 99 |
| [60] | C.T. Images | 3D-DNN, COVID-19Net | AUC 0.86 Sensitivity of 79.35% and specificity of 71.43% | China | 95 |
| [61] | C.T. Images | CNN, Multi-task learning, self-supervised learning, DenseNet-169 | Accuracy of 0.89 and AUC 0.90 | China | 175 |
| [44] | C.T. Images | Five classifiers, Decision tree, k-nearest neighbor, naïve Bayes, support vector machine, Proposed COVIDiag model | Accuracy of 91.4% sensitivity of 93.24%, and specificity of 90.32% | Iran | 7 |
| [36] | X-ray Images | CNN, CoroNet | Overall Accuracy 89.6% | India | 159 |
| [37] | X-ray Images | CNN VGG16 | Average accuracy 0.97% | Italy | 69 |
| [11] | X-ray Images | CNN DarkCovidNet | Accuracy of 98.08% | Turkey | 417 |

Table 2. *Cont.*

| Reference | Test Type | ML/DL Techniques | Prediction Results | Country | Cited by No of Papers |
|-----------|-----------|------------------|-------------------|---------|----------------------|
| [40] | CT Images | 3-D CNN ResNet-18 | Overall Accuracy 86.7% | China | 443 |
| [3] | X-ray Images | CNN, MobileNet v2 | Accuracy of 96.78% Sensitive 98.66% Specificity 96.46% | Greece | 480 |
| [45] | X-ray Images | D.N.N., VGG-19 ResNet-50, COVID-Net | Accuracy of 93.3% | Canada | 558 |
| [46] | X-ray Images | CNN, RestNet50 + SVM | Accuracy of 95.33% | India | 250 |
| [47] | X-ray Images | D-CNN, VGG19, DenseNet201 | Classification with F1-scores of 0.89 and 0.91 | Egypt | 251 |
| [68] | X-ray Images | CNN, COV19-ResNet, COV19-CNNet | Accuracy of 97.61% | Turkey | 1 |
| [48] | X-ray Images | ResNet50, DenseNet201, Inception-v3, and Xception | AUC 0.996, Overall sensitivity of 0.94 | USA | 2 |
| [51] | X-ray Images | CNN, Bayesian ResNet50V2 | Accuracy of 89.92% | UK | 104 |
| [55] | X-ray Images | DCNN, CheXNet + DenseNet-201 | Accuracy of 99.7% precision, sensitivity of 99.7% and specificity is 99.55% | Qatar | 161 |
| [57] | X-ray Images | CNN, Classification Grad-CAM | AUC 95.13%, the sensitivity of 90%, and specificity of 87.84% | China | 161 |
| [64] | X-ray Images | CNN, COVID-ResNet | Accuracy of 96.23% | USA | 125 |
| [65] | X-ray Images | D-CNN, DeTraC | Accuracy of 95.12% Sensitivity of 97.91% and Specificity of 91.87% | Egypt | 161 |
| [66] | X-ray Images | D.N.N., Deep COVID Explainer | Positive Predictive Value 96.12% and recall of 94.3% | Germany | 34 |
| [67] | X-ray Images | VGG-16 VGG-19 | Accuracy of 87.49% | Australia | 5 |
| [38] | X-ray Images | CNN, AlexNet, GoogleNet, SqueezeNet | Overall accuracy 99% | Saudi Arabia | 4 |
| [69] | X-ray Images | Classification Models K.N.N., ANN, D.T., SVM, | Overall accuracy of 93.41% | India | 8 |

**Table 2.** *Cont.*

| Reference | Test Type | ML/DL Techniques | Prediction Results | Country | Cited by No of Papers |
|---|---|---|---|---|---|
| [35] | Chest X-ray CT Images | CNN, RestNet50, Inception V3, Inception-RestNetV2 | Accuracy of 98% | Turkey | 408 |
| [53] | Chest X-ray CT Images | D-CNN, A.I. system | A.U.C. of 97.91% The sensitivity of 90.19% and specificity of 95.76% | China | 97 |
| [56] | Chest X-ray CT Images | Simple 2-D CNN Model and pre-trained AlexNet with transfer learning | Accuracy of 98% on X-ray images and 94.1% on C.T. images | Iraq | 80 |
| [62] | Chest X-ray CT Images | D-CNN, Resnet50, MobileNet_V2, Inception_Resnet_V2 | Accuracy of 96.61% | Morocco | 39 |
| [58] | RT-PCR | Support Vector Machine, Random Forests, Neural Networks, Logistic Regression | AUC 0.847, Sensitivity 0.67, Specificity 0.85 | Brazil | 31 |
| [59] | RT-PCR | Support Vector Machine, K.N.N., Decision Tree, Random Forest | Accuracy of 80% | China | 147 |
| [70] | Clinical Blood Test | Random Forests (R.F.) | Accuracy of 0.9512, Sensitivity 0.9697, and Specificity 0.9595 | China | 27 |
| [71] | Clinical Blood Test | CRISP-DM, Random Forest, Deep Neural Network, Extreme gradient Boosting Machine (XGBoost) | A.U.C. of 0.97, Sensitivity 81.9%, and specificity of 97.9% | Switzerland | 5 |
| [72] | Clinical Blood Test | Six Classification Model BayesNet, logistic-regression, lazy-classifier, meta-classifier, classification via regression, decision-tree (J48) | Accuracy of 84.24% | Turkey | 0 |

In RT-PCR, samples collected from a person's body, such as nose or throat, are investigated to detect the presence of the virus. The reliability of RT-PCR is not suitable owing to a high false rate, due to which it cannot give accurate results [73]. The availability of RT-PCR is relatively short around the world. Moreover, it includes a lengthy procedure for detection and is quite expensive. RT-PCR based model revealed with the least accuracy of 80% in recent research [59] and the maximum A.U.C. of 0.84 for another analysis [58]. The clinical blood test-based model was found with the lowest accuracy of 84.24% in the previous

study [72] and the highest A.U.C. of 0.97 for another study [66]. Comparatively, the clinical blood test-based model obtained the highest A.U.C. of 0.97 [71] than RT-PCR-based models.

In the C.T. scan method, a final chest image is captured by combining the images taken from various angles, while a C.T. scan requires a short time, but it is quite expensive. It can be observed that the C.T. image-based model obtained a minimum A.U.C. of 0.86 in the previous study [60] and maximum accuracy of 99.96% as per the investigation [50].

X-rays are used to create images of the chest. An X-ray is economically affordable in most areas. Therefore, most of the medical experts utilized X-ray images instead of C.T. images. An X-ray image-based model was observed with the lowest accuracy of 86.7% in a previous study [40] and the highest accuracy of 99.7% for another study [55].

Currently, ML/DL has played a significant role in boosting many CAD systems' diagnostic efficiency for various medical applications such as diagnosing and detecting different pulmonary diseases. Recent studies have shown that DL techniques have proven to be very efficient for CAD systems in radiography [48]. Hence, C.T. scan and X-ray both are widely used imaging methodologies for detecting various diseases and COVID-19. While X-rays need less data memory, processing time requires a low radiation dose than a C.T. scan [74]. Thus, CAD systems present vital solutions to boost and support radiologist workflow in predicting COVID-19 using low-dose X-ray images and overcoming limitations.

The literature has investigated that some researchers employed C.T. and X-ray image-based combined models, which were observed with the lowest accuracy of 96.61% in a previous study [62] and the highest accuracy of 98% for another study [35]. It can be stated that radiological X-ray image-based models worked better than the C.T. image-based models by examining Table 2 because the highest accuracy of X-ray-based models is 99.7% by using D.C.N.N. [55] to classify COVID-19. Thus, the C.T. and X-ray images are optimal for detecting coronavirus, but medical experts are required for the RT-PCR test, C.T. scan, and X-ray techniques.

It was observed that CNN and D.N.N. were the most considerable classification techniques for detecting COVID-19, followed by SVM, Random Forest followed by K-NN, and L.S.T.M. Moreover, compared to other ML/DL techniques, CNN was the most widely utilized classifier for the diagnosis of COVID-19. D.C.N.N. was the most accurate for the detection of COVID-19, but its usage was relatively short. Hence it needs to be explored further in future research for the detection and diagnosis of COVID-19. The study revealed that ML/DL-based approaches can significantly promote intelligent diagnosis systems, which are promising for healthcare professionals to make fast and reliable detection of the virus. It will also eliminate the manual flaws during the diagnosis by physicians and radiologists. Moreover, it will be a step towards time-efficient and accurate diagnoses to facilitate both hospitals and the patients.

## 4. State-of-the-Art COVID-19 Detection Using Anal Swab-Based Diagnosis

The outbreak of COVID-19 has adversely impacted the whole world with a massive death rate. In late December 2019, China was the epidemic center just before the spring festival [75]. However, with this perspective, the Chinese government has taken strict precautions to overcome the spread of the virus. Some of the best strategies deployed by China to combat this pandemic are given below.

1. Stop traveling from in and out of China to control the transmission.
2. Developed quarantine centers for suspected cases to get the best treatment.
3. China developed mobile apps for tracking suspected, confirmed cases, and interaction with individuals having COVID-19 symptoms.
4. Develop public awareness regarding self-protection, epidemiologic investigation, cleaning, and disinfecting the environment.
5. The government installed many intelligent based systems to monitor the public temperature, such as airports, metro stations, hospitals, communities.

There are different criteria for suspected, confirmed, asymptomatic, mild, and critical cases. All these cases were diagnosed and cured by following management criteria to

diagnose COVID-19. Moreover, China's current positive cases have dropped to a few or even zero [76]. However, the pandemic situation goes on critically in other countries. Noticeably, China's policy implemented special preventive and diagnostic measures to control the deadly virus transmission and spread with more individuals testing. Oropharyngeal, nasopharyngeal swabs are primarily used for nucleic acid sampling [77].

According to Li Tong (Deputy Chief Physician of the department of respiratory and infectious disease of Beijing You'an Hospital), oropharyngeal swab tests are widely used and most appropriate for large-scale sampling. Nevertheless, the nasopharyngeal swab test accuracy is higher than the oropharyngeal swab tests, but the nasopharyngeal swab test is not convenient as oropharyngeal, and a person feels uncomfortable while sampling. On the flip side, China has started anal swab testing in Beijing to enhance infected people's detection rate and minimize unpredictable diagnoses. The sample collection method for anal swabs is to insert a cotton-tipped swab about 1-2 inches into the rectum to detect the virus. However, the sample collection of the anal swab is not so appropriate as of oropharyngeal swab. Therefore, anal swab sample collection is limited to high-risk areas such as quarantine centers. It was observed that the virus lives a long time in the anus than the upper respiratory tract. Therefore, the anal swab tests remain positive in infected cases after giving negative results in the oropharyngeal swab [7]. Noticeably, it is concluded that anal swab tests could accurately identify the virus than any other swab tests. The significant purpose of anal swabs is to increase the detection rate and increase accuracy in identifying COVID-19 positive cases.

## 5. Challenges and Future Research Directions

Several challenges are required to be addressed in applying ML/DL techniques to detect COVID-19. Various kinds of diagnosis and detection strategies have been developed to control the spread of COVID-19 based pandemic, but the knowledge related to the applications of various ML/DL techniques to detect the virus is still insufficient. ML/DL approaches need a massive amount of data for computational models to discover and achieve information that is very limited in the case of COVID-19. The ML/DL techniques can only provide better results and valuable solutions to deal with the pandemic if a considerable amount of clinical data and medical images are available. To solve the issue, researchers need to work on appropriate ML/DL techniques that can provide better results using small datasets [78] such as L.S.T.M. [79].

Moreover, Multi-Criteria Decision Analysis (M.C.D.A.) [80] can be considered a practical solution that enables decision-making to deal with the issues related to COVID-19. As Li et al. [26] specify, radiology images give optimal results in differentiating coronavirus from other kinds of viral pneumonia, but it still needs to be more efficient to recognize imaging features accurately. In some cases, the presence of noisy, unclear, and inaccurate datasets can lead to misclassification of radiological images (C.T. and X-ray), which can be a significant difficulty in diagnosing COVID. Many complications hinder the training of ML/DL algorithms, such as data redundancy and missing values. Henceforth, researchers utilized different kinds of datasets to improve the training of algorithms to detect the virus. It will enable the radiologists and doctors to diagnose the C.T. and X-ray images more efficiently and promote awareness of viruses. Therefore, a standard framework is needed for future research to assess and analyze these issues. ML can be applied for preparator and feedback activities to tackle worldwide critical situations. For example, ML applications can be used to develop the most efficient robotic and automatic setup for sanitation, distributing food, medicine, and taking care of patients in hospitals. In the computer and biomedical engineering domain, ML has been partially utilized to find out the novel drug compounds against coronavirus [81,82].

Moreover, the usage of data science applications like ultrasound and Magnetic Resource Imaging (M.R.I.) for the detection of COVID-19 was limited. The ultrasound scans are as useful as chest C.T. scans, but these are not reported to be utilized to detect COVID-19. Some researchers [83] employed M.R.I to detect COVID-19, but due to the shortage of

data, the M.R.I. technique's feasibility and effectiveness are still unexplained. Generating well-defined datasets for the effective use of these methods to detect COVID-19 is another challenge. Moreover, upcoming research must be focused on the classification of COVID-19 by different symptoms of illness (such as fever, cough, congestion, sore throat, runny nose, diarrhea, and breathing difficulty) for precise and fast diagnosis in chronic patients (organ transplant). It is essential to utilize advanced ML/DL techniques to accurately evaluate infected patients and the corresponding death rate in the current scenario. One of the critical issues to be solved is the lack of interest by medical experts in examining medical images and clinical data. Furthermore, the manual classification of COVID-19 is a complicated and time-consuming process that demands the utilization of intelligent supervised ML/DL algorithms [84,85].

## 6. Conclusions

The global outbreak of novel coronavirus has affected millions of lives and demands various cost-effective diagnostic tests about the presence of COVID-19 infection. Fortunately, the current era of advanced technologies with ML/DL techniques has improved various medical aspects of human life and detect chronic and contagious diseases. There is a need for robust research to overcome the spread of this deadly virus by using ML-/DL-based intelligent models and taking preventive measures. In the recent systematic review, the analysis of ML/DL algorithms reported in the literature related to COVID-19 prediction, classification, and detection strategies has been presented. Several researchers have utilized X-ray, CT images, RT-PCR, and clinical blood data to assist the prognosis and anomalies of COVID-19. The analysis was performed to select appropriate ML/DL techniques to predict and diagnose the virus using radiological and clinical datasets. It is revealed that the highest accuracy of various ML/DL techniques such as CNN, D.N.N, SVM, K-NN, and R.F is 99%, 99.7%, 99.68%, 93.41%, and 95%, respectively, to detect COVID-19. These ML/DL techniques have achieved astounding performance results in every domain, along with medical research and radiology. However, DL has become dominant in various complicated tasks such as image classification and detection. Being familiar with these models' key advantages will assist radiologist's diagnosis research and develop an automated medical diagnosis decision support system for medical health experts. Finally, it can be concluded that ML and DL techniques played a significant role in the prediction, classification, screening, and minimizing the spread of the COVID-19 pandemic.

**Author Contributions:** Conceptualization, A.R. and H.X.; methodology, A.R., H.X., and M.A.I.; validation, A.R., H.X., M.A.I., and I.A.; formal analysis, M.A.I.; investigation, A.R., H.X., and M.A.I.; resources, H.X.; writing-original draft, A.R.; writing-review and editing, A.R., H.X., M.A.I., and I.A.; visualization, A.R., and I.A.; supervision, H.X. All authors have read and agreed to the published version of the manuscript.

**Funding:** This work was supported by the China Scholarship Council, China, grant number 201907005026.

**Institutional Review Board Statement:** Not applicable.

**Informed Consent Statement:** Not applicable.

**Conflicts of Interest:** The authors report no conflict of interest.

**Abbreviations**

The following abbreviations are used in this manuscript.

| | |
|---|---|
| ML | Machine Learning |
| MLT | Machine Learning Techniques |
| CT | Computer Tomography |
| CAD | Computer-Aided Diagnosis |
| CNN | Convolutional Neural Network |
| D.N.N | Deep Neural Network |
| DL | Deep Learning |
| DT | Decision Tree |
| RF | Random Forest |
| SVM | Support Vector Machine |
| SARS | Severe Acute Respiratory Syndrome |
| K-NN | K-Nearest Neighbors |
| L.S.T.M. | Long Short-Term Memory |
| SD | Science Direct |
| W.O.S | Web of Science |
| X-ray | X-radiation |
| M.R.I | Magnetic Resource Imaging |
| MERS | Middle East Respiratory syndrome |
| RT-PCR | Reverse Transcription Polymerase Chain Reaction |
| A.U.C. | Area under the Receiver Operating Characteristic Curve |

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
