# Peer review of "COVID-19 Detection Empowered with Machine Learning and Deep Learning Techniques: A Systematic Review"

_applsci, doi:10.3390/app11083414_

Round 1

Reviewer 1 Report

COVID-19 has infected 223 countries and caused 2.6 million deaths worldwide (at the 8 time of writing this article), and the death rate is increasing continuously. Early diagnosis of 9 COVID patients is a critical challenge for medical practitioners, governments, organizations, and 10 countries to overcome the rapid spread of the deadly virus in any geographical area. In this situation, the previous epidemic evidence on Machine Learning (ML) and Deep Learning (DL) techniques encouraged the researchers to play a significant role in detecting COVID-19. Similarly, the 1rising scope of ML/DL methodologies in the medical domain also advocates its significant role in COVID-19 detection. This systematic review presents ML and DL techniques practiced in this era to predict, diagnose, classify, and detect the coronavirus. In this study, the data was retrieved from three prevalent full-text archives, i.e., Science Direct, Web of Science, and PubMed, using the search code strategy on March 16, 2021. The authors should address the following comments to improve the presentation of their manuscript.

(1As this paper is related to computer-aided diagnosis (CAD), so authors should discuss few lines about CAD in the introduction section and include few recent references reacted CAD. That will be more interesting for the readers.

(2) Based on the comment (1), the authors should appropriate revise the related work, i.e.

Karar, M. E., Hemdan, E. E. D., & Shouman, M. A. (2021). Cascaded deep learning classifiers for computer-aided diagnosis of COVID-19 and pneumonia diseases in X-ray scans. Complex & Intelligent Systems, 7(1), 235-247.

Hizukuri, A., Nakayama, R., Nara, M., Suzuki, M. and Namba, K., 2021. Computer-Aided Diagnosis Scheme for Distinguishing Between Benign and Malignant Masses on Breast DCE-MRI Images Using Deep Convolutional Neural Network with Bayesian Optimization. Journal of Digital Imaging, 34(1), pp.116-123. Khan, M. A., & Kim, Y. (2021). Cardiac arrhythmia disease classification using LSTM deep learning approach. CMC-COMPUTERS MATERIALS & CONTINUA, 67(1), 427-443.Kim, Soo-Yeon, Yunhee Choi, Eun-Kyung Kim, Boo-Kyung Han, Jung Hyun Yoon, Ji Soo Choi, and Jung Min Chang. "Deep learning-based computer-aided diagnosis in screening breast ultrasound to reduce false-positive diagnoses." Scientific Reports 11, no. 1 (2021): 1-11.

English should be revised and few extra spaces should be remove from the current version.

Author Response

Dear Editor and Reviewers,

Thank you for allowing us to submit a revised draft of our manuscript titled "COVID-19 Detection Empowered with Machine Learning and Deep Learning Techniques: A Systematic Review" to Applied Sciences Journal. We appreciate the time and effort that you and the reviewers have dedicated to providing valuable feedback on our manuscript. We are grateful to the reviewers for their insightful comments on our paper. We have highlighted the changes within the manuscript. Here we have attached the file attachment. Please check it.

Reviewer 2 Report

In this article, the authors aim to analyse different ML/DL techniques that have been recently published related to the detection of coronavirus and to consider upcoming research challenges.
The article is well written and minor details on English grammar require review. Moreover, this work is well organized, with proper structure and readability. A clear objective is set, the proposed research is marked and supported in the literature. The bibliography is sufficient and well given.
I think this article has good potential but, before being considered ready for publication, some aspects need to be clarified and improved.
1)First of all, in the "Review Methodology" section, the authors need to elaborate, why the traditional ML/DL classifiers are selected. The models need to be explained more technical with more details from ML/DL technical points of view.
2) The "Results and Discussion" section needs enrichment. There is no clear presentation of the results and their commentary. The authors did not explain how these methods used. Please, explain them clearly. Try to make a more coherent, accurate and focused presentation.
3) In the conclusion section, the authors have not highlighted any numeric information related to accuracy and other advantages of the proposed solutions.
4)I recommend the authors include more references from MDPI.

Author Response

(The authors gave the same response as above.)
